

# Deep learning-based anomaly detection using one-dimensional convolutional neural networks (1D CNN) in machine centers (MCT) and computer numerical control (CNC) machines

Ali Athar[1], Md Ariful Islam Mozumder[1], Abdullah[2], Sikandar Ali[1] and Hee-Cheol Kim[1]

[1] Digital Anti-aging Healthcare, Inje University, GIMHAE, Gyeongsangnam-do, Republic of South Korea
[2] James Cook University of North Queensland, Queensland, Australia

## ABSTRACT

Computer numerical control (CNC) and machine center (MCT) machines are mechanical devices that manipulate different tools using computer programming as inputs. Predicting failures in CNC and MCT machines before their actual failure time is crucial to reduce maintenance costs and increase productivity. This study is centered around a novel deep learning-based model using a 1D convolutional neural network (CNN) for early fault detection in MCT machines. We collected sensor-based data from CNC/MCT machines and applied various preprocessing techniques to prepare the dataset. Our experimental results demonstrate that the 1D-CNN model achieves a higher accuracy of 91.57% compared to traditional machine learning classifiers and other deep learning models, including Random Forest (RF) at 89.71%, multi-layer perceptron (MLP) at 87.45%, XGBoost at 89.67%, logistic regression (LR) at 75.93%, support vector machine (SVM) at 75.96%, K-nearest neighbors (KNN) at 82.93%, decision tree at 88.36%, naïve Bayes at 68.31%, long short-term memory (LSTM) at 90.80%, and a hybrid 1D CNN + LSTM model at 88.51%. Moreover, our proposed 1D CNN model outperformed all other mentioned models in precision, recall, and F-1 scores, with 91.87%, 91.57%, and 91.63%, respectively. These findings highlight the efficacy of the 1D CNN model in providing optimal performance with an MCT machine's dataset, making it particularly suitable for small manufacturing companies seeking to automate early fault detection and classification in CNC and MCT machines. This approach enhances productivity and aids in proactive maintenance and safety measures, demonstrating its potential to revolutionize the manufacturing industry.

## INTRODUCTION

Managing failures in manufacturing processes is necessary in the manufacturing industry. Manufacturing machines can cause a loss in the manufacturing process, sometimes

Corresponding authors
Ali Athar, ali.athar1401@gmail.com
Hee-Cheol Kim, heeki@inje.ac.kr

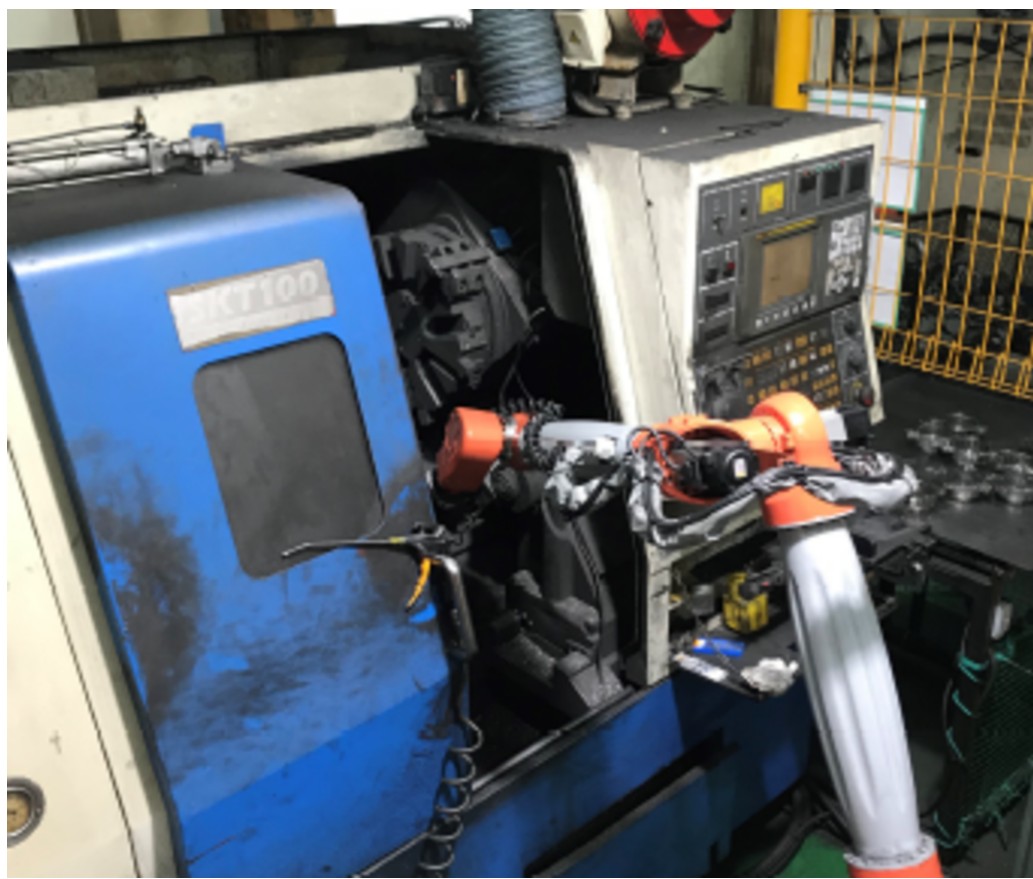

**Figure 1  Overview of machine center machine.** Image source credit: vertical 5-Axis CNC machining centres - Mills CNC.

because of their high complexity or default. An anomaly during production is not beneficial to a company because it affects its overall success. In the case of a manufacturing machine's failure, an alternative solution should be used, such as a spare system in an emergency. This will be very costly and will require regular maintenance, even if they remain unused. Therefore, manufacturing industries seek cost-effective and efficient ways to handle machine failures. Predictive maintenance is one of the options used for this purpose; it provides a promising path to machine failures and avoids anomalies in the manufacturing process (*Barlow & Hunter, 1960*; *Hoppenstedt et al., 2018*; *Saxena et al., 2008*).

MCT and CNC machines are used in manufacturing processes to manufacture different metals. These machines are handled by coding on a PC *via* a programming application. Various kinds of commands are generated through coding to perform various operations. Multiple data types are produced from these machines, such as temperature, vibration, power, spindle, and noise. Many researchers have proved that from all kinds of data, vibration, spindle, energy, and noise data are the main reasons for machine failure (*Jiang et al., 2019*). Figure 1 shows a picture of an MCT machine (*Kim et al., 2022*).

Moreover, the trend toward machine learning produces an interrogation of the possibility of predicting manufacturing machine failure, thus detecting anomalies in the manufacturing process. Although the manufacturing industry has witnessed some technical developments to address these defects and failures, their practical application remains a challenging task for many reasons (*Hoppenstedt et al., 2018*; *Kammerer et al., 2019*). Based on these considerations, our study presents a real-world case conducted in cooperation with a manufacturing company. The company uses computer numerical control (CNC) and machine centers (MCT) to drill metals. Several defects during these manufacturing processes can be observed, reducing the devices' efficiency. Thus, for a company, defect detection and prevention remain of utmost importance as they will increase the productivity and efficiency of its machines. However, defect detection in these machines remains a complex task that requires considerable time and effort. Thus far, many manufacturing companies have employed selected choice experts to detect anomalies based on their long-term experience. However, such experts are generally expensive and sometimes ineffective. Moreover, these experts took too much time to observe several machines. Based on sensor data collected from MCT machines, we aim to minimize manual decisions regarding defect detection and improve productivity (*Feng et al., 2019*; *Sadaf, Athar & Azam, 2016*; *Schlechtendahl et al., 2015*; *Vodenčarević et al., 2011*).

The company's real-world data are related to an MCT machine that monitors records during the drilling of the metals. The process must be continuously monitored to reduce costly downtime. Continuous monitoring generates large amounts of sensor data related to several MCT machine components.

This study aimed to assess and predict the early failure of MCT machines. Training our proposed one-dimensional convolutional neural network (1D-CNN) base model that can use sensor-based data collected from a manufacturing machine capable of predicting defects will significantly contribute to our research. Our experimental results demonstrate the 1D CNN's superior performance in anomaly detection for MCT machines. The model's ability to accurately identify anomalies, even with limited training data, underscores its practical value in real-world manufacturing settings. This innovation enhances the fault diagnosis process and contributes to the broader adoption of deep learning techniques in the industrial sector. In the past, researchers have proposed different techniques to predict early defects in these machines. Still, there are some drawbacks, such as some of them using only vibration data and some of them using only spindle values in their research. Also, some of the researchers did not perform feature engineering techniques. Moreover, there is a need for comprehensive evaluation metrics that go beyond accuracy to include precision, recall, F1-score, and statistical significance tests. Considering these factors, we have tried to fill the research gap using a combination of three types of datasets obtained from the CNC and MCT machine, and we also performed features engineering techniques to enhance the performance of our proposed model. This 1D-CNN-based deep-learning (*Du, 2018*; *Lei, Pan & Huang, 2019*) model assists small manufacturing companies in early predicting anomalies in CNC and MCT machines. Furthermore, we have conducted paired t-tests to statistically evaluate the performance differences between the proposed 1D CNN model and other traditional machine learning and deep learning models. This evaluation

not only determines our model's superior performance in terms of precision, recall, and F1 score but also confirms the statistical significance of the observed improvements, thereby reinforcing the reliability and robustness of our approach.

A summary of our work is provided below:

- Applying a state-of-the-art 1D CNN deep learning model on a custom sensor dataset from a private manufacturing company's CNC and MCT machine.
- We are pioneers in using a combination of spindle datasets, power, and vibration data, as it is easy for small companies to record spindle values in MCT and CNC machines.
- After applying different preprocessing techniques, we extracted seven features, a unique combination that improved the accuracy of our proposed model. No one else used this combination in research on manufacturing machines.
- To address the class imbalance problem, we utilized the class weight technique, which helped balance both effective and defective classes.
- We trained a state-of-the-art 1D CNN deep learning model and compared its performance with eight other machine learning classifiers, including Random Forest (RF), XGBoost, multi-layer perceptron, logistic regression (LR), K-nearest neighbor (KNN), Support vector machine (SVM), decision tree, and naïve Bayes.
- Furthermore, the proposed model's performance was compared with other deep learning models, such as long short-term memory (LSTM) and the hybrid CNN and LSTM (CNN + LSTM) model.
- Finally, we have conducted paired t-tests to statistically evaluate the performance differences between the proposed 1D CNN model and all other models.

The remainder of this paper is organized as follows:

Related work is discussed in 'Related Work', and 'Materials & Methods' describes the methods and materials used. 'Proposed Algorithm' briefly explains the proposed algorithm. The results are given in 'Results', and finally, 'Conclusions and Future Work' provides conclusions and future directions.

## RELATED WORK

Various machine and deep learning algorithms have been employed with notable success in anomaly detection within manufacturing processes. However, the 1D-CNN utilized in our research presents distinct advantages over these methods, particularly in CNC and MCT machines and other manufacturing applications.

One-dimensional convolutional neural networks (1D CNNs) are particularly well-suited for analyzing sequential data, such as the time-series data generated by sensors in MCT and CNC machines. Unlike traditional machine learning methods, which often rely on handcrafted features, 1D CNNs automatically learn the temporal dependencies and patterns directly from the raw data. This capability is crucial for capturing the dynamic behavior of manufacturing processes, where anomalies may manifest as subtle temporal variations in the sensor readings (*Hussain, Ali & Kim, 2022a*; *Yao et al., 2017*).

Machine-learning algorithms detect anomalies using different datasets produced by manufacturing machines. Recently, *Pittino et al. (2020)* investigated anomaly detection

methods in in-production manufacturing machines. They demonstrated that the nature of the available data, featuring any anomaly, was vital for algorithmic choice. In this study, automatic anomaly detection methods using statistical machine learning and control charts are developed and deployed effectively in an in-production environment. They used support vector machine-based fault diagnostics for the induction motors. Their study was based on vibration, current signals, and critical fault detection features. These features were fed into the SVM as inputs to diagnose faults. The obtained model was tested under several conditions and proven highly effective (*Pittino et al., 2020*). *Chen et al. (2019)* proposed a method for predicting transient stability in power systems using the XGBoost machine learning model. By simulating the New England 10-machine 39-bus power system with PSASP, the study generates time-domain data to train the model. The approach leverages feature selection techniques to enhance model performance and emphasizes interpretability through decision rules and feature importance scores. Their proposed XGBoost model outperforms other machine learning classifiers, including RF, DT, SVM, and neural network (NN), in accuracy and efficiency (*Chen et al., 2019*). Using NASA hypertext transfer protocol (HTTP) log datasets, *Henriques et al. (2020)* combined XGBoost and K-means algorithms to detect anomalies in the dataset. The built model can be used for forensic and compliance auditing analyses of security management (*Henriques et al., 2020*). *Gao, Ma & Yang (2002)* used classifiers to assign anomaly labels to a set of k-length windows obtained from a training dataset. The classifier can detect anomalous windows that are either ignored or assigned a defect label (*Gao, Ma & Yang, 2002*). For a sequence of sensor data collected from an operational aircraft system (*Budalakoti et al., 2006*; *Budalakoti, Srivastava & Otey, 2008*; *Sriastava, 2005*) Anomaly detection techniques were applied to handle sequence-based problem formulation and identify faulty functional runs. *Selvaraj & Min (2023)* retrofitted a CNC machine to monitor energy consumption and detect defects in real-time. Using data from a power meter, they extracted different features across time, frequency, and time-frequency domains. They used five machine learning classifiers for anomaly detection: DTs, RFs, k-NN, SVM, and Bagging classifiers. Their proposed model achieved high accuracy and was deployed using AWS for real-time monitoring (*Selvaraj & Min, 2023*).

Deep learning technology has risen quickly and achieved significant success in several domains, including object detection, intelligent robots, saliency detection, sound event detection for innovative city safety, UAV blade problem diagnostics, and parking garage sound event detection. Researchers have used deep learning methods to detect product defects and have improved productivity and quality (*Yang et al., 2020*). Intelligent manufacturing, a cornerstone of Industry 4.0, heavily relies on machine vision-based surface defect detection to ensure product quality. CNNs play a pivotal role in this context by providing robust and automated solutions for detecting surface defects across diverse industries, thus supporting the goals of Industry 4.0 (*Khanam et al., 2024*). Tao et al. research makes two contributions: The first is a multitask 1D-CNN system for wire defect detection, which combines object detection and object categorization networks. Second, they suggested that deep learning should be applied in this area first. The real-world dataset of an industrial production plant shows how well the proposed approach identifies defects

(*Tao et al., 2018*). *Kim et al. (2022)* proposed an abnormal data classification method for intelligent devices in CNC machines. They proposed a hybrid method that combined singular spectrum analysis (SSA) techniques, which were very efficient in decreasing the noise in time-series data, with convolutional autoencoder (CAE), which performed very well in their time-series data (*Kim et al., 2022*). *Jiang et al. (2022)* presented a novel approach to counter-error prediction using deep learning and reinforcement. Gubernatorov et al. proposed a method to detect faults in CNC machine gearboxes using vibration data. They developed a CNN-based VGG model that distinguishes between healthy (OK) and faulty (NOK) processes during high-speed machine operations. Their proposed research integrates their model into an edge-to-cloud architecture for automated fault detection and data annotation (*Gubernatorov & Gavrilenkov, 2024*). *Chengyang, Sitong & Wansheng (2021)* proposed a method in which the axial and radial thermal errors of horizontal and vertical spindles are modeled using a deep-learning convolutional neural network (CNN). Their model blends thermocouple data with the thermal image to accurately reflect the temperature field of the spindle (*Chengyang, Sitong & Wansheng, 2021*). *Zhao et al. (2021)* proposed an improved target defect detection algorithm for steel surfaces. They improved the target detection accuracy on the NUE-DET dataset by using the R-CNN algorithm (*Zhao et al., 2021*).

Anomaly detection to natural language processing deep learning has been effectively used in time series modeling and prediction problems in various application domains. Recurrent neural networks (RNNs) were developed to address the limitations of conventional feed-forward neural networks for modeling sequences. However, LSTM networks have recently become popular for sequence and sequence-to-sequence learning because of the well-known vanishing gradients problem of RNN (*Brunelli et al., 2019*). They used the internal data of CNC machines to track errors in the feeding axis and modeled it as a nonlinear auto-regressor LSTM (*Jiang et al., 2022*). Masci et al. proposed a multiscale pooling technique that can detect flaws on steel surfaces and capture photos of various sizes as inputs. Their proposed method outperformed many traditional methods available in the market (*Masci et al., 2013*). *Sharma, Chawla & Ram (2020)* developed an automatic CNC program for operating multiple types of holes in CNC machines using support vector machines and restricted Boltzmann machines (RBM) with a deep belief network (DBN). *Li, Bedi & Melek (2023)* proposed an LSTM Autoencoder-based method to detect anomalies in CNC machines. They utilized transfer learning for this purpose. Their initial trained model could distinguish between stable and unstable cutting conditions based on vibration data obtained from the accelerometer. They introduced additional layers and re-trained their model to enhance the detection accuracy. Experimental results validate the ability of their proposed model to detect anomalies, with performance metrics showing promise for industrial application in the automated manufacturing industry (*Li, Bedi & Melek, 2023*).

Table 1 provides a comprehensive comparative analysis of several research studies, summarizing the machine used, dataset, model, and contribution of each study.

# MATERIALS & METHODS

We followed different steps in this study. Each is explained below:

**Table 1  Summary of various studies on anomaly detection techniques and models applied in the manufacturing industry.**

| Paper | Machin used | Dataset | Proposed model | Feature used | Summary & Findings |
|---|---|---|---|---|---|
| *Pittino et al. (2020)* | Induction motors | Vibration current signal | SVM | Average standard deviation Maximum Minimum Kurtosis | The study demonstrates the effectiveness of combining control charts and anomaly classification algorithms to detect anomalies in the DEU. |
| *Chen et al. (2019)* | 10-machine 39-Bus Power System | Generators' Speeds Rotor angles Power | XGBoost | Rotor Angle Kinetic Energy | Utility of XGBoost in advancing transient stability prediction in power systems, offering a robust framework that balances computational efficiency with interpretability, thus enhancing its applicability in real-time operations and emergency response scenarios. |
| *Henriques et al. (2020)* | NASA Kennedy Space Centre | NASA HTTP logs | XGBoost K-means | IP address Page Operation Method response. | The proposed method combines k-means clustering and XGBoost classification to detect anomalies in large log datasets efficiently. It leverages parallel computing to handle big data scenarios, providing a scalable solution with interpretable results. |
| *Selvaraj & Min (2023)* | CNC/MCT Machines | Power consumption data | Decision Tree Random Forest K-NN SVM Bagging | Time domain Frequency Domain Time-Frequency Domain | fault detection in CNC machines by leveraging power consumption data and deploying advanced machine learning models |
| *Kim et al. (2022)* | CNC Machine | Vibration Data | Convolutional Autoencoder | N/A | This paper demonstrates a robust approach to abnormal data classification in CNC machines, leveraging advanced deep learning techniques and efficient data preprocessing methods to achieve high accuracy in real-world applications. |
| *Gubernatorov & Gavrilenkov (2024)* | CNC Machine | Vibration Data | VGG | Time Domain Frequency Domain | Detects CNC machine gearbox faults using vibration data and a VGG model, leveraging raw time-domain and frequency-domain features. |
| *Brunelli et al. (2019)* | CNC machines | CAD files containing information about the dimensions and features of the holes | Restricted Boltzmann Machine (RBM) SVM | Tool Characteristics | The study concludes that RBM with DBN is more suitable for large datasets and offers better accuracy and classification results, while SVM is more suitable for small datasets |
| *Jiang et al. (2022)* | CNC/MCT Machine | Contour errors | NAR-LSTM Time-Series Deep Q-Network | Reference positions of X & Y axes. Velocity Velocity Jerk | Effectively integrates deep learning and reinforcement learning to model and compensate for errors in CNC machining |
| *Li, Bedi & Melek (2023)* | CNC Machine | Vibration Data | LSTM | Frequency domain | Proposed an LSTM autoencoder network to detect anomalies in CNC machines. |

## Study design and subjects

This was a cross-sectional and multicenter observational study carried out at Inje University. The institutional review board approved this study of Kyungnam University, and all the participants provided consent to participate in this study.

## Data collection and data preprocessing

We used datasets from a single MCT machine, identified as number 0K43A16211, provided by the manufacturing company. To ensure a controlled environment where the machine's operational parameters and external conditions remained consistent, we utilized a one-day labeled dataset from July 7, 2021. Also, its representative nature, including defective and effective classes, made it sufficient for our model's training and validation. It demonstrates the model's capability to detect anomalies with limited data, reflecting real-world manufacturing scenarios where extensive historical data may not always be available.

The company collected the data using three sensors: current sensors for measuring power, tachometers for spindle speed, and accelerometers for vibration levels. The sensors recorded data at an average rate of 10 Hz per second, providing a detailed and continuous data stream. This frequency sampling allowed for a thorough and granular analysis of the machine's operational parameters.

Our dataset comprises 26,126 records, divided into effective (normal) and defective (abnormal) classes. Table 2 illustrates the division of the two classes.

Data processing and cleaning are essential steps in data science and play a vital role in increasing the overall performance of AI models. In the data extraction process, meaningless data were removed to achieve better results. In our dataset, unnecessary attributes not required for further processing were removed.

We used a one-hot (*Potdar, Pardawala & Pai, 2017*) representation to label defective and effective datasets. '0' is used to represent defective, and '1' is used to describe an effective dataset.

Our dataset contained some missing values, particularly in the power feature set. Incorrect handling of these missing values yields a less accurate model and can decrease the overall performance of the machine-learning models (*Rahm & Do, 2000*). We used medians to fill the empty values in our dataset. Finally, we used StandardScaler (*Athar et al., 2021*; *Ghosh et al., 2021*) to transform all values in a column to a range of mean 0 and standard deviation 1. Figure 2 shows the steps involved in the data preprocessing.

## Experimental procedure

We processed and analyzed our dataset using a system with the following specifications: Windows 10 Pro-64-bit, 2.50 GHz Core i7-11700 processor, 16 GB RAM, Python, and TensorFlow 2.7.0, manufactured by HP and sourced from Gimhae, South Korea.

## Statistical feature engineering

The model's performance depends on the features selected after the data preprocessing. Identifying useful features and removing unnecessary features from large amounts of data are difficult. Overfitting in an efficient machine learning model can be avoided by selecting

**Table 2  Overview of data distribution.**

| Class | Total data | No. of values for power | No. of values for spindle | No. of values for vibration |
|---|---|---|---|---|
| Effective | 9,811 | 9,811 | 9,811 | 9,811 |
| Defective | 16,315 | 16,315 | 16,315 | 16,315 |
| Total | 26,126 | 26,126 | 26,126 | 26,126 |

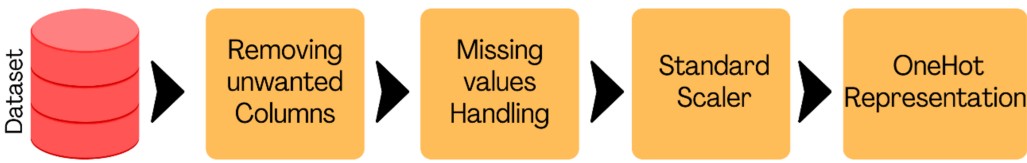

**Figure 2  Data preprocessing steps.**

a smaller number of excellent feature subsets (*Hussain et al., 2022b*). Seven more different features for all three basic features were selected, including mean, standard deviation, minimum, maximum, energy measure, interquartile range, and skewness (*Kankar, Sharma & Harsha, 2011*; *Rafiee, Rafiee & Tse, 2010*). With the selection of these seven features, the 1D CNN model gave a maximum performance, as has been proved already in the research study done by *Hussain, Ali & Kim (2022a)*.

## Class weight technique

Different techniques are used for data balancing, including the class weight. Using this technique, we develop a weighting algorithm to calculate the loss function. Depending on the imbalanced dataset, weights were assigned to both the majority and minority classes. A threshold value should be defined to maintain balance among different classes. This threshold value helps to increase or decrease class weights, so ultimately, it will prevent the bias of the model toward any particular class (*Hussain, Ali & Kim, 2022a*). The class weight is expressed in Eq. (1):

$$w^i = \frac{n\_instances}{n\_classes \times n\_instancesi} \tag{1}$$

where,
wi = wight of each class and I = Each Class
$n\_instances$ = Total number of instances in the whole dataset
$n\_classes$ = Total number of classes
$n\_instancesi$ = Total number of rows in each class.

Table 3 shows an implementation of the class weight technique in our study. We calculated the class weight using Eq. (1) and assigned 1.33 weights for the minority class *i.e.,* effective class. This gave more weight to the minority class than the majority class, *i.e.,* defective. Resultantly, it helped to address the data imbalance issue in our dataset and improved the performance of our model.

**Table 3  Class weights.**

| Class | Weight |
| --- | --- |
| Defective (0) | 0.66 |
| Effective (1) | 1.33 |

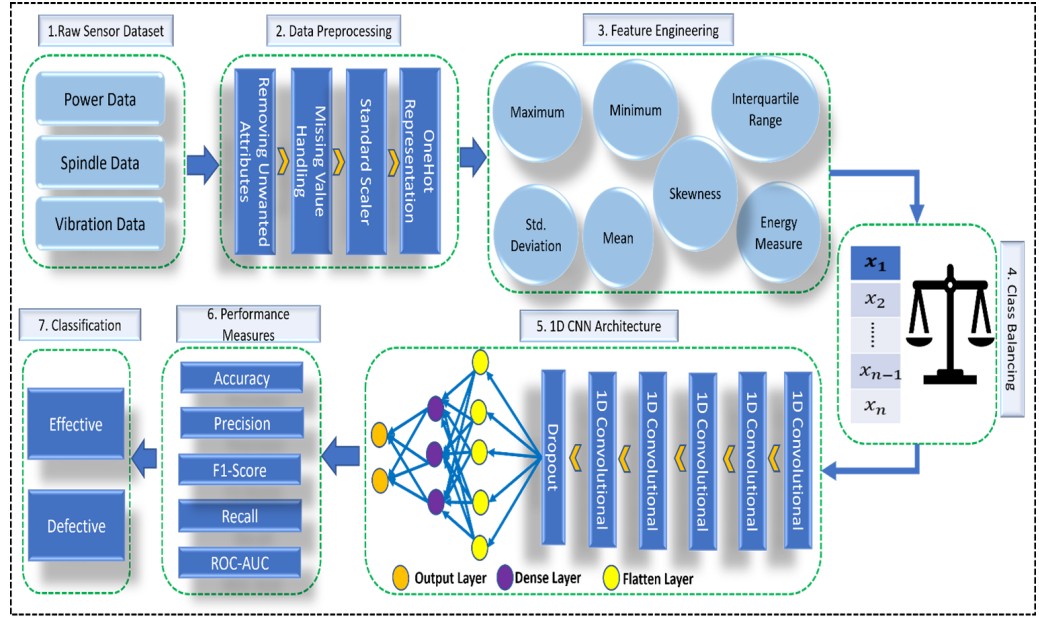

**Figure 3  Overview of proposed architecture.** Created using PowerPoint.

### Complete architecture of proposed methodology

A complete picture of our work is given in Fig. 3.

## PROPOSED ALGORITHM

We used three different types of sensor data to develop our prediction algorithm. Different types of data are power, spindle, and vibration. In addition, we applied the other preprocessing techniques discussed in 'Proposed Algorithm' to make our data ready for use in our proposed algorithm. Using this algorithm, we predicted the anomaly of the CNC and MCT machines. The function and Training of our proposed model are explained by the pseudo-code given in Table 4.

### Data format and measurements

We created a matrix of $24 \times 30 = 720$ feature vectors, where 24 denotes the total number of attributes, and 30 is the number of rows in each window. Figure 4 explains this process, later used for model training and testing while predicting the results.

To avoid robustness and overfitting of results, the entire dataset was divided into 80% for training, 10% for validation, and 10% for testing. This division was implemented for all eight machine learning and three deep learning models. This data division simplifies

Athar et al. (2024), *PeerJ Comput. Sci.*, DOI 10.7717/peerj-cs.2389

**Table 4  Pseudo-code of function and training of 1D-CNN.**

| (a) Function of 1D CNN Model | (b) Training of 1D CNN Model |
|---|---|
| *Code:* | *Code:* |

```
FUNCTION CNN_MODEL(x)
        weight ← DEFINEWEIGHTS
        biases ← DEFINEBIASES
        x ← RESHAPE(X)
        conv1 ← RELU_ACTIVATION_FUNC(Conv1D(x))
        conv2 ← RELU_ACTIVATION_FUNC(Conv1D(x))
        conv3 ← RELU_ACTIVATION_FUNC(Conv1D(x))
        conv4 ← RELU_ACTIVATION_FUNC(Conv1D(x))
        conv5 ← RELU_ACTIVATION_FUNC(Conv1D(x))
        conv5 ← DROPOUT(0.1)
        conv5 ← FLATTEN(0.1)
        conv5 ← DROPOUT(0.1)
        dense1 ← RELU_ACTIVATION_FUNC()
        dense2 ← RELU_ACTIVATION_FUNC()
        dense3 ← RELU_ACTIVATION_FUNC()
        RETURN output
END FUNCTION
```

```
FEATURE ← {"POWER","SPINDLE","VIBRATION",
"POWER_MIN","POWER_MAX","POWER_IR","POWER_SD",
"POWER_MEAN","POWER_SK","POWER_EM","SPINDLE_MIN",
"SPINDLE_MAX","SPINDLE_IR","SPINDLE_SD",
"SPINDLE_MEAN","SPINDLE_SK","SPINDLE_EM","VIBRATION_MIN",
"VIBRATION_MAX","VIBRATION_IR","VIBRATION_SD",
"VIBRATION_MEAN","VIBRATION_SK","VIBRATION_EM",}
CLASSES ← {"EFFECTIVE","DEFECTIVE"}
X ← dataset{FEATURES}.values
Y ← dataset{CLASSES}.values
TRAIN_DATA, TEST_DATA, VALID_DATA TEST_TRAIN_SPLIT(X,Y,0.2,0.5)
BATCHSIZE ← 4
LOSS binary_crossentropy, learning_rate ← 1e07, Epoches ← 250
1DCNN_MODEL.compile ← (LOSS, learning_rate)
1DCNN_MODEL.train ← (TRAIN_DATA, EPOCHS, BATCH_SIZE, VALID_DATA)
```
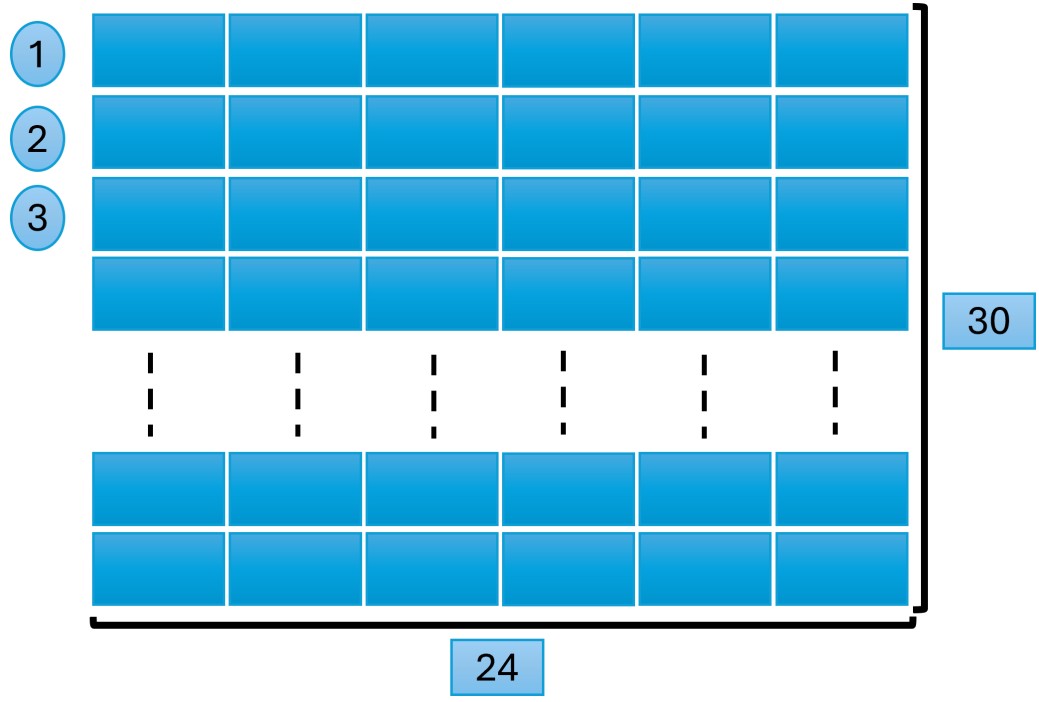

**Figure 4 Window size and data scanning.**

the analysis and ensures that our model's performance metrics are more straightforwardly interpretable.

## Architecture of 1D CNN

In our study, we used deep learning techniques to predict the effectiveness of our sensor data. Among these deep learning techniques, the 1D CNN model is very efficient for a small dataset, as our results prove.

Deep learning has an additional benefit over traditional AI, which consists of classification and feature extraction (*Bibbò & Carotenuto, 2021*) We can automatically extract the most relevant and essential features using these deep learning models. The 1D CNN is another highly effective artificial neural network (ANN) model used for feature extraction and classification.

## Input layer

The data with dimensions (30×24) were fed into the input layer of the 1D CNN. Where 30 is the window size, and 24 represents the number of features.

## Convolution layer

Convolutional operations were applied to the input data with a stride size 1. The five convolutional layers use a kernel size of 4. The filter sizes in these four convolutional layers were 4, 4, 4, 4, and 8.

### Dropout

Dropout helps avoid overfitting the model. In our dropout layers of the 1D CNN model, we used a dropout value of 0.5.

### Output

The performance of the CNN model is highly dependent on the activation function. In our model, we used a rectified linear unit (ReLU) as an activation function in two dense layers in our experiments to obtain a better result, and the sigmoid function was used as an activation function in the third dense layer of our proposed model. We also applied Stochastic gradient descent (SGD) in our model, and a learning rate of 0.0001 was applied. Finally, _binary_crossentropy_ was used as a loss function to measure the total loss between the predicted and actual numbers.

We conducted a series of preliminary experiments to fine-tune selected parameters and ensure that the chosen configuration offers a balance between model performance and training efficiency.

Table 5 gives a summary of these fine-tuned parameters.

## RESULTS

In this section, we discuss our study's experimental results. We compared the performance of our proposed 1D CNN model with eight machine learning models—random forest, XGB, MLP, LR, SVM, KNN, DT, and naïve Bayes—and one deep learning model, the LSTM. Additionally, we evaluated a hybrid model combining 1D CNN and LSTM (1D CNN+LSTM).

### Performance measures

We utilized various performance metrics to evaluate the models, including accuracy, precision, recall, F1-score, and the AUC-ROC curve. The following equations define these metrics:

$$Accuracy = \frac{TP + TN}{TP + TN + FP + FN} \tag{2}$$

$$Precision = \frac{TP}{TP + FP} \tag{3}$$

$$Recall = \frac{TP}{TP + FN} \tag{4}$$

$$F1 - Score = \frac{2 \times Precision \times Recall}{Precision + Recall}. \tag{5}$$

Table 6 highlights the superior performance of our proposed 1D CNN model compared to ten other models, which include eight machine learning models and two deep learning models. This table provides a comprehensive comparative analysis of all the models evaluated.

**Table 5  Parameters and their descriptions.**

| Parameter | Value | Description |
|---|---|---|
| Initial learning rate | 0.0001 | Starting learning rate for training. |
| Learning rate schedule | Exponential decay | Learning rate decay method. |
| Optimizer | SGD with nesterov momentum | Optimization algorithm used for training. |
| Momentum | 0.9 | Momentum parameter for SGD. |
| Epochs | 250 | Number of training iterations. |
| Batch size | 4 | Number of samples per gradient update. |
| Conv1D layers | 6 Layers | The number of convolutional layers. |
| Filters | 4, 4, 4, 8, 8 | Number of filters in each Conv1D layer. |
| Kernel size | 2 | Size of the convolutional kernels. |
| Activation function | ReLU | Activation function for Conv1D layers. |
| Dense layers | 3 Layers | Number of dense layers. |
| Activation | ReLU, ReLU, Sigmoid | Activation functions for the dense layers. |
| Loss function | Binary crossentropy | Loss function used for training. |

**Table 6  Precision, recall and F1-score of individual classes in all models.**

| | | Precision (%) | Recall | F1-Score |
|---|---|---|---|---|
| Logistic regression | Defective (0) | 81.32 | 79.78 | 80.54 |
| | Effective (1) | 67.39 | 69.52 | 68.44 |
| Support vector classifier | Defective (0) | 81.14 | 80.15 | 80.64 |
| | Effective (1) | 67.63 | 69.01 | 68.31 |
| K-Nearest neighbors | Defective (0) | 87.11 | 85.29 | 86.19 |
| | Effective (1) | 76.35 | 79.00 | 77.66 |
| Decision tree | Defective (0) | 90.49 | 90.93 | 90.71 |
| | Effective (1) | 84.79 | 84.10 | 84.44 |
| Naive bayes | Defective (0) | 97.74 | 50.43 | 66.53 |
| | Effective (1) | 54.32 | 98.06 | 69.91 |
| Random forest | Defective (0) | 91.84 | 91.67 | 91.75 |
| | Effective (1) | 86.18 | 86.44 | 86.31 |
| XGBoost | Defective (0) | 92.99 | 90.26 | 91.60 |
| | Effective (1) | 84.55 | 88.69 | 86.57 |
| MLP | Defective (0) | 90.15 | 89.71 | 89.93 |
| | Effective (1) | 83.01 | 83.69 | 83.35 |
| LSTM | Defective (0) | 93.21 | 92.07 | 92.64 |
| | Effective (1) | 86.87 | 88.66 | 87.76 |
| Hybrid | Defective (0) | 96.53 | 84.76 | 90.26 |
| | Effective (1) | 78.63 | 94.85 | 85.98 |
| 1D-CNN (the proposed model) | Defective (0) | 95.51 | 90.85 | 93.13 |
| | Effective (1) | 85.71 | 92.78 | 89.11 |

**Table 7  Overall result summary of all models.**

|  | Precision | Recall (%) | F1-Score (%) | Accuracy (%) |
|---|---|---|---|---|
| RF | 89.71 | 89.71 | 89.71 | 89.71 |
| XGB | 89.82 | 89.67 | 89.71 | 89.67 |
| MLP | 87.47 | 87.45 | 87.47 | 87.45 |
| LR | 76.09 | 75.93 | 76.00 | 75.93 |
| SVM | 76.07 | 75.96 | 76.01 | 75.96 |
| K-NN | 83.07 | 82.93 | 82.99 | 82.93 |
| DT | 88.35 | 88.36 | 88.36 | 88.36 |
| NB | 81.44 | 68.31 | 67.80 | 68.31 |
| LSTM | 90.85 | 90.80 | 90.82 | 90.80 |
| 1D CNN+ LSTM | 89.88 | 88.51 | 88.67 | 88.51 |
| **1D-CNN (the proposed model)** | **91.87** | **91.57** | **91.63** | **91.57** |

Table 7 summarizes the average accuracy achieved by the 1D CNN and the ten other models. Our proposed model performed very well on testing data and got maximum accuracy of 91.57%, 91.87% Precision, 91.57% Recall, and 91.63% F1-Score measures compared to other machine learning and deep learning models.

Additionally, it presents the confusion matrix, which is crucial for understanding the performance of the machine learning models. The confusion matrix includes four metrics: true positive (TP), true negative (TN), false positive (FP), and false negative (FN). True positives and true negatives represent correctly classified instances, while false positives and false negatives represent type I and type II errors, respectively. For a detailed performance breakdown, the normalized confusion matrices of each model are illustrated in Table 8.

### Auc-Roc curve

The AUC-ROC curve is a crucial performance criterion for evaluating deep learning and machine learning models. A higher AUC indicates a more robust and superior model for differentiating between classes. Figure 5A presents a comparison of the 1D-CNN model with naïve Bayes, XGBoost, LR, LSTM, and a hybrid model of LSTM and 1D-CNN. Similarly, the AUC-ROC curve in Fig. 5B compares the 1D-CNN with the other remaining five models. These images clearly demonstrate that our proposed 1D-CNN model significantly outperforms the other ten models.

### Paired t-tests

We conducted paired t-tests to assess the statistical significance of the differences in accuracies between the 1D-CNN model and each of the other models. The results are summarized in Table 9. This table presents the results of paired t-tests assessing the statistical significance of differences in peak accuracy between the 1D-CNN and other models. The t-statistic and $p$-value for each comparison are provided to determine whether the differences are statistically significant. All comparisons show statistically significant differences ($p < 0.05$), indicating that the CNN model outperforms the other models' accuracy.

**Table 8  Confusion matrix of each model.**

| Model | | Positive | Negative |
|---|---|---|---|
| Random forest (RF) | Positive | 0.92 | 0.08 |
| | Negative | 0.14 | 0.86 |
| XGBoost | Positive | 0.90 | 0.10 |
| | Negative | 0.11 | 0.89 |
| MLP | Positive | 0.90 | 0.10 |
| | Negative | 0.16 | 0.84 |
| LR | Positive | 0.80 | 0.20 |
| | Negative | 0.30 | 0.70 |
| SVM | Positive | 0.80 | 0.20 |
| | Negative | 0.31 | 0.69 |
| K-NN | Positive | 0.85 | 0.15 |
| | Negative | 0.21 | 0.79 |
| DT | Positive | 0.91 | 0.09 |
| | Negative | 0.16 | 0.84 |
| NB | Positive | 0.50 | 0.50 |
| | Negative | 0.02 | 0.98 |
| LSTM | Positive | 0.92 | 0.08 |
| | Negative | 0.11 | 0.89 |
| Hybrid model | Positive | 0.85 | 0.15 |
| | Negative | 0.05 | 0.95 |
| 1D CNN | Positive | 0.91 | 0.09 |
| | Negative | 0.07 | 0.93 |

# CONCLUSIONS AND FUTURE WORK

While several researchers have focused on manufacturing, research on MCT and CNC machines remains sparse. This study addresses this gap by applying a 1D-CNN deep learning model for anomaly detection in MCT/CNC machines. Our results demonstrate that the 1D-CNN model effectively predicts MCT machines' operational status and defects, outperforming eight traditional machine learning models and one deep learning model. Our model achieved the highest accuracy and outperformed a hybrid 1D CNN and LSTM model, even with a small dataset.

Although our study was limited to data collected for a single day, the 1D CNN's strong performance suggests its potential for effective anomaly detection with limited data. Future work will use data from multiple days and varying operational conditions to further validate our findings to ensure the model's robustness and adaptability. Exploring more advanced deep learning techniques may enhance MCT machine productivity.

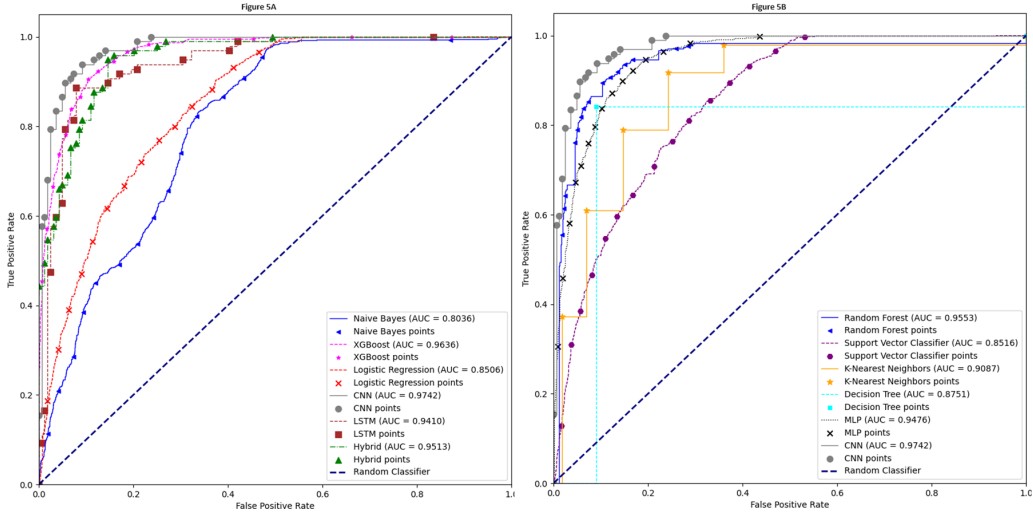
**Figure 5** AUC-ROC curve.

**Table 9** T-test summary.

| Model comparison | t-statistic | p-value | Statistical significance |
|---|---|---|---|
| 1D-CNN *vs* LSTM | 9.4632 | 2.0721e−20 | Significant ($p < 0.05$) |
| 1D-CNN *vs* Hybrid (LSTM+ 1D CNN) | 36.8330 | 2.7106e−188 | Significant ($p < 0.05$) |
| 1D-CNN *vs* Naive Bayes | 377.6534 | 0.0000e+00 | Significant ($p < 0.05$) |
| 1D-CNN *vs* Random Forest | 353.1712 | 0.0000e+00 | Significant ($p < 0.05$) |
| 1D-CNN *vs* Logistic regression | 345.0091 | 0.0000e+00 | Significant ($p < 0.05$) |
| 1D-CNN *vs* Support vector classifier | 323.3016 | 0.0000e+00 | Significant ($p < 0.05$) |
| 1D-CNN *vs* K-Nearest neighbors | 331.7052 | 0.0000e+00 | Significant ($p < 0.05$) |
| 1D-CNN *vs* Decision tree | 313.1611 | 0.0000e+00 | Significant ($p < 0.05$) |
| 1D-CNN *vs* XGBoost | 349.6871 | 0.0000e+00 | Significant ($p < 0.05$) |
| 1D-CNN *vs* MLP | 342.2846 | 0.0000e+00 | Significant ($p < 0.05$) |

### Funding

This research was supported by the MSIT (Ministry of Science ICT), Korea, under the National Program for Excellence in SW, supervised by the IITP (Institute of Information & Communications Technology Planning & Evaluation) in 2022 (2022-0-01091, 1711175863). The funders had no role in study design, data collection and analysis, decision to publish, or preparation of the manuscript.

### Grant Disclosures

The following grant information was disclosed by the authors:

The MSIT (Ministry of Science ICT), Korea, under the National Program for Excellence in SW, supervised by the IITP (Institute of Information & Communications Technology Planning & Evaluation) in 2022: 2022-0-01091, 1711175863.

## Competing Interests

The authors declare there are no competing interests.

## Author Contributions

- Ali Athar conceived and designed the experiments, performed the experiments, analyzed the data, performed the computation work, prepared figures and/or tables, authored or reviewed drafts of the article, and approved the final draft.
- Md Ariful Islam Mozumder conceived and designed the experiments, performed the experiments, analyzed the data, prepared figures and/or tables, and approved the final draft.
- Abdullah conceived and designed the experiments, performed the experiments, analyzed the data, performed the computation work, authored or reviewed drafts of the article, and approved the final draft.
- Sikandar Ali conceived and designed the experiments, prepared figures and/or tables, and approved the final draft.
- Hee-Cheol Kim conceived and designed the experiments, performed the experiments, analyzed the data, performed the computation work, authored or reviewed drafts of the article, supervision, Funding acquisition, and approved the final draft.

## Data Availability

The data and code are available at Zenodo: ATHAR, A., & Kim, H.-C. (2024). Manufactoring Dataset of MCT machine [Data set]. Zenodo. https://doi.org/10.5281/zenodo.13141031.

## Supplemental Information

Supplemental information for this article can be found online at http://dx.doi.org/10.7717/peerj-cs.2389#supplemental-information.

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
