# Peer review of "Deep learning-based anomaly detection using one-dimensional convolutional neural networks (1D CNN) in machine centers (MCT) and computer numerical control (CNC) machines"

_PeerJ Computer Science, doi:10.7717/peerj-cs.2389_

## Round 0.1 · original submission · Major Revisions

Dear authors,
You are advised to critically respond to all comments point by point when preparing a new version of the manuscript and while preparing for the rebuttal letter. Please address all the comments/suggestions provided by the reviewers.

Kind regards,
PCoelho

Reviewer 1 ·

Basic reporting

All comments have been added in detail to the 4th section called additional comments.

Experimental design

All comments have been added in detail to the 4th section called additional comments.

Validity of the findings

All comments have been added in detail to the 4th section called additional comments.

Additional comments

Review Report for PeerJ Computer Science
(Deep learning-based anomaly detection and classification using 1D CNN in MCT & CNC Machines)

1. Within the scope of the study, various classification processes were carried out with deep learning and machine learning using the sensor-based dataset obtained from micro compression testing (MCT) and computer numerical controls (CNC) machines.

2. In the introduction, the importance of MCT and CNC machines, the necessity of early fault detection and the contribution of the study to the literature are clearly mentioned.

3. In the Related works section, machine learning and deep learning-based models and ensemble method approaches in the literature regarding anomaly detection in machines are explained. In this section, it is recommended to add a literature table consisting of columns such as "machines used, dataset, proposed model, results, pros and cons" of the studies in the literature in order to more clearly emphasize the importance of the subject and its difference from the literature. Thus, it is expected that the study will have the potential to compare the literature and bring its originality to the fore.

4. In the study, a normal/abnormal two-class dataset of three different types, spindle, vibration and power, obtained from the machines at Kyungnam University, was used. It is very valuable that the data set used was obtained specifically for the study. Although the preprocessing operations performed on the dataset are important, it is recommended to add information about the dataset such as "total amount of data, amount of data for each class, amount of data for each type, amount of data for each type in each class" by detailing it in a table.

5. In classification problems, how the dataset distribution is determined is very important. It was stated that within the scope of this study, the distribution was determined as 80% for training and 20% for testing. How were these dataset distribution percentages determined? It is recommended to explain more clearly how the distribution is determined, as the results may vary depending on the test dataset, especially after network training, in the randomly selected dataset. Normally, cross-validation is frequently used for accurate analysis and reliability of the results. Explain why cross-validation was not preferred in this study and/or how the dataset distribution percentages were determined.

6. More detailed information should be given about the parameters used for the CNN model proposed in the study (such as learning rate, optimizer, epoch), and it should be explained on what basis they were determined and whether different experiments were made.

7. It was stated that the proposed CNN model was compared with random forest and XGBoost, which are machine learning models. Although the results obtained show that CNN gives better results, it should be explained what determines these two machine learning models chosen for comparison. Since there are many different machine learning models in the literature, it is recommended to compare them with several different models.

8. Comparison of CNN model results with two machine learning models, multi-layer perceptron, deep learning-based hybrid model and long short-term memory model, which is a recurrent neural network, increased the depth of the study.

9. When we look at the evaluation metrics in terms of analysis of the results, it is observed that all important metrics such as ROC curve, AUC score, confusion matrix, precision, recall, accuracy are obtained. Considering both the metric type and the results, it is understood that it has a sufficient level.

As a result, although the study is important in terms of the originality of the dataset and the proposed model, it is recommended that all parts mentioned above be explained.

Reviewer 2 ·

Basic reporting

The topic is well-described and structured. The Title of the paper should be modified to “Deep learning-based anomaly detection using 1D CNN in MCT & CNC Machines.
There is no need to include the word classification in the title since Detection and classification virtually mean the same thing. Moreover, the critical analysis aspect is weaker, as the literature review fails to highlight the key gaps at the machine learning level. For example, it does not identify research gaps in the methodology of the reviewed literature at the preprocessing and classification levels.

Experimental design

Methods are well described. Paper is in the scope of the journal

Validity of the findings

Originality is stated. Conclusions should be supported by results. Authors added limitations and directions or further research

Additional comments

There are many typos and grammatical errors within the manuscript that need to be addressed.

Reviewer 3 ·

Basic reporting

Grammar and Clarity: A thorough proofread is essential. Consider using a professional editing service to ensure the manuscript meets publication standards.
Literature Review: The current review is superficial. Provide a more focused discussion on existing work using 1D CNNs for anomaly detection, particularly in the context of MCT machines. Highlight the limitations of these methods and how your study addresses these gaps.
Structure: Consider combining redundant sections like "Data Collection" and "Data Preprocessing" for improved flow.
Figures and Tables: While the presence of figures is noted, their quality is lacking. Ensure figures are clear, high-resolution, and accompanied by descriptive captions.

Experimental design

Research Question: The research question is clear but lacks novelty. Emphasize the unique contribution of the 1D CNN application for anomaly detection in MCT machines compared to existing approaches.
Data Details: The methodology section is thin on details. Provide justification for using data from a single day and address potential limitations arising from this choice. Include information about the sensor sampling rate and how the data features were chosen.

Validity of the findings

Impact and Novelty: The manuscript fails to adequately demonstrate the impact or novelty of the proposed approach. Highlight the specific advantages of 1D CNNs for anomaly detection in MCT machines compared to existing methods. Explain how this work benefits the field of anomaly detection in manufacturing.
Replication: Encourage replication by acknowledging the limitations of the small dataset and discussing how a larger dataset might influence the results.
Statistical Significance: The current analysis focuses solely on peak accuracy. Conduct and report statistical tests to assess the significance of the differences observed between the proposed model and other approaches.
Conclusions: The conclusions should be more specific and directly linked to the study's findings. Avoid generic statements about deep learning models. Focus on the specific contributions of the 1D CNN application and its performance for anomaly detection in MCT machines.

Additional comments

* Clarity: The abstract states CNCs and MCT machines are electrochemical devices. This might be a minor inaccuracy. It would be better to clarify if they are electrically powered or involve some electrochemical processes.
* Focus on Results: In the abstract, while mentioning high accuracy (91.57%), it would be helpful to mention other relevant metrics like precision, recall, or F1 score for a more comprehensive picture.
* Focus & Conciseness: The abstract can be shortened by removing redundant information (e.g., "electrochemical devices"). Emphasize key findings and avoid generic statements about deep learning models.
* Focus & Conciseness: The introduction can be more concise by removing redundant information (e.g., details about company location and specific data types being collected).
* Focus & Conciseness: The Related work section is overly long and includes unnecessary details about ensemble methods (e.g., bagging, voting). Focus on the most relevant related work using CNNs and LSTMs for anomaly detection in MCT machines specifically.

---

## Round 0.2 · accepted · Accept

Dear authors, we are pleased to verify that you meet the reviewer's valuable feedback to improve your research.

Thank you for considering PeerJ Computer Science and submitting your work.

Reviewer 1 ·

Basic reporting

All comments have been added in detail to the last section.

Experimental design

All comments have been added in detail to the last section.

Validity of the findings

All comments have been added in detail to the last section.

Additional comments

Review Report for PeerJ Computer Science
(Deep learning-based anomaly detection using one-dimensional convolutional neural networks (1D CNN) in Machine Centers (MCT) and Computer Numerical Control (CNC) machines)

Thank you for the revision. The responses to the reviewer comments are generally sufficient. In addition, the changes made to the paper are appropriate. For this reason, I recommend that the paper be accepted. I wish the authors success in their future papers.

Reviewer 2 ·

Basic reporting

The authors have addressed all my comments and the manuscript has comprehensively improved. Also, other reviewers comments have been well-addressed

Experimental design

All comments addressed

Validity of the findings

The findings from the research have validity now

Additional comments

The paper has been improved now

Reviewer 3 ·

Basic reporting

No comment

Experimental design

No comment

Validity of the findings

No comment